# Semi-supervised novelty detection using ensembles with regularized disagreement

**Alexandru Țifrea**[1]    **Eric Stavarache**[1]    **Fanny Yang**[1]

[1]Department of Computer Science, ETH Zurich, Switzerland

## Abstract

Deep neural networks often predict samples with high confidence even when they come from unseen classes and should instead be flagged for expert evaluation. Current novelty detection algorithms cannot reliably identify such near OOD points unless they have access to labeled data that is similar to these novel samples. In this paper, we develop a new ensemble-based procedure for *semi-supervised novelty detection* (SSND) that successfully leverages a mixture of unlabeled ID and novel-class samples to achieve good detection performance. In particular, we show how to achieve disagreement only on OOD data using early stopping regularization. While we prove this fact for a simple data distribution, our extensive experiments suggest that it holds true for more complex scenarios: our approach significantly outperforms state-of-the-art SSND methods on standard image data sets (SVHN/CIFAR-10/CIFAR-100) and medical image data sets with only a negligible increase in computation cost.

## 1 INTRODUCTION

Despite achieving great in-distribution (ID) prediction performance, deep neural networks (DNN) often have trouble dealing with test samples that are out-of-distribution (OOD), i.e. test inputs that are unlike the data seen during training. In particular, DNNs often make incorrect predictions with high confidence when new unseen classes emerge over time (e.g. undiscovered bacteria [Ren et al., 2019], new diseases [Katsamenis et al., 2020]). Instead, we would like to automatically *detect* such novel samples and bring them to the attention of human experts.

Consider, for instance, a hospital with a severe shortage of qualified personnel. To make up for the lack of doctors, the

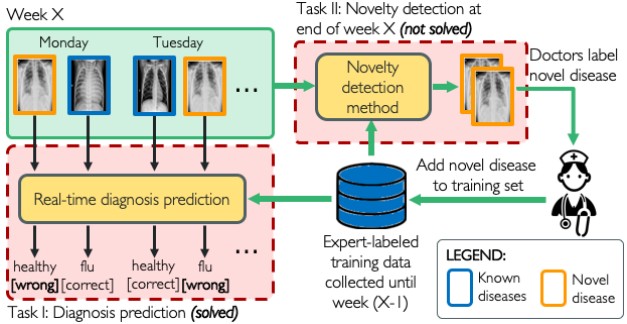

Figure 1: Novelty detection is challenging since X-rays of novel diseases are remarkably similar to known conditions. The unlabeled batch of inference-time data can be used to adapt a semi-supervised novelty detection approach to emerging novel diseases.

hospital would like to use an automated system for real-time diagnosis from X-ray images (Task I) and a novelty detection system, which can run at the end of each week, to detect outbreaks of novel disease variants (Task II) (see Figure 1). In particular, the detection algorithm can be fine-tuned weekly with the unlabeled batch of data collected during the respective week.

While the experts are examining the peculiar X-rays over the course of the next week, the novelty detection model helps to collect more instances of the same new condition and can request human review for these patients. The human experts can then label these images and include them in the labeled training set to update both the diagnostic prediction and the novelty detection systems. This process repeats each week and enables both diagnostic and novelty detection models to adjust to new emerging diseases.

Note that, in this example, the novelties are a particular kind of out-of-distribution samples with two properties. First, several novel-class samples may appear in the unlabeled batch at the end of a week, e.g. a contagious disease will lead to several people in a small area to be infected. This situation is different from cases when outliers are assumed

*Accepted for the 38th Conference on Uncertainty in Artificial Intelligence (UAI 2022).*

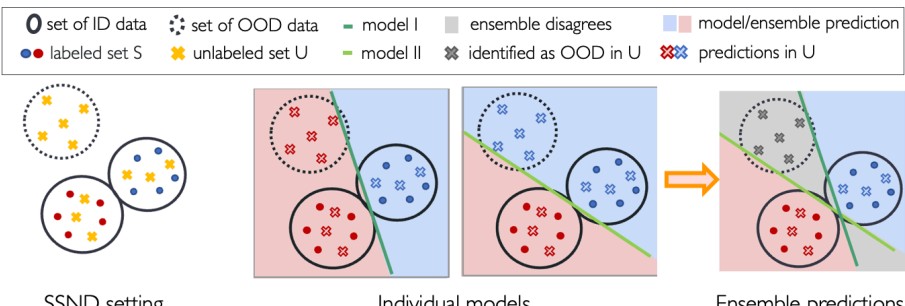

Figure 2: **Left:** Sketch of the SSND setting. **Middle and Right:** Novelty detection with a diverse ensemble.

to be singular, e.g. anomaly detection problems. Second, the novel-class samples share many features in common with the ID data, and only differ from known classes in certain minute details. For instance, both ID and OOD samples are frontal chest X-rays, with the OOD samples showing distinctive signs of a pneumonia caused by a new virus. In what follows, we use the terms *novelty detection* and *OOD samples* to refer to data with these characteristics.

Automated diagnostic prediction systems (Task I) can already often have satisfactory performance [Calli et al., 2021]. In contrast, novelty detection (Task II) still poses a challenging problem in these scenarios. Many prior approaches can be used for semi-supervised novelty detection (SSND), when a batch of unlabeled data that may contain OOD samples is available, like in Figure 1.[1] However, all of these methods fail to detect novel-class data when used with complex models, like neural networks.

Despite showing great success on simple benchmarks like SVHN vs CIFAR10, SOTA unsupervised OOD detection methods perform poorly on near OOD data [Winkens et al., 2020] where OOD inputs are similar to the training samples. Furthermore, even though unlabeled data can benefit novelty detection Scott and Blanchard [2009], existing SSND methods for deep neural networks [Kiryo et al., 2017, Guo et al., 2020, Zhang et al., 2020, Yu and Aizawa, 2019] cannot improve upon unsupervised methods on near OOD data sets. Even methods that violate fundamental OOD detection assumptions by using known test OOD data for hyperparameter tuning [Liang et al., 2018, Lee et al., 2018, Yu and Aizawa, 2019] fail to work on challenging novelty detection tasks. Finally, large pretrained models seem to solve near OOD detection [Fort et al., 2021], but they only work for extremely specific OOD data sets (see Section 5 for details).

This situation naturally raises the following question:

*Can we improve semi-supervised novelty detection for neural networks?*

---

[1] We use the same definition of SSND as the survey by Bulusu et al. [2020], whereas some works use the term to refer to supervised [Gornitz et al., 2013, Daniel et al., 2019, Ruff et al., 2020] or unsupervised ND [Song et al., 2017, Akçay et al., 2018] according to our taxonomy in Section 5.

In this paper, we introduce a new method that successfully leverages unlabeled data to obtain diverse ensembles for novelty detection. Our contributions are as follows:

- We propose to find Ensembles with Regularized Disagreement (ERD), that is, disagreement only on OOD data. Our algorithm produces ensembles just diverse enough to be used for novelty detection with a disagreement test statistic (Section 2).

- We prove that training with early stopping leads to regularized disagreement, for data that satisfies certain simplifying assumptions (Section 3).

- We show experimentally that ERD significantly outperforms existing methods on novelty detection tasks derived from standard image data sets, as well as on medical image benchmarks (Section 4).

## 2 PROPOSED METHOD

In this section we first introduce our proposed method to obtain Ensembles with Regularized Disagreement (ERD) and describe how they can be used for novelty detection.

### 2.1 TRAINING ENSEMBLES WITH REGULARIZED DISAGREEMENT (ERD)

Recall from Figure 1 that we have access to both a labeled training set $S = \{(x_i, y_i)\}_{i=1}^n \sim P$, with covariates $x_i \in \mathcal{X}_{ID}$ and discrete labels $y_i \in \mathcal{Y}$, and an unlabeled set $U$, which contains both ID and unknown OOD samples. Moreover, we initialize the models of the ensemble using the weights of a predictor with good in-distribution performance, pretrained on $S$. In the scenarios we consider, such a well-performing pretrained classifier is readily available, as it solves Task I in Figure 1.

The entire training procedure is described in Algorithm 1. For training a single model in the ensemble, we assign a label $c \in \mathcal{Y}$ to all the unlabeled points in $U$, resulting in the $c$-labeled set that we denote as $(U, c) := \{(x, c) : x \in U\}$. We then fine-tune a classifier $f_c$ on the union $S \cup (U, c)$ of the correctly-labeled training set $S$, and the unlabeled set $(U, c)$. In particular, we choose an early stopping time at which

**Algorithm 1:** Obtaining ERD ensemble via early stopping

**Input :** Train set $S$, ID Validation set $V$, Unlabeled set $U$, Model $\tilde{f}$ pretrained on $S$, Ensemble size $K$

**Result:** ERD ensemble $\{f_{y_i}\}_{i=1}^K$

Sample $K$ different labels $\{y_1, ..., y_K\}$ from $\mathcal{Y}$

**for** $c \leftarrow \{y_1, ..., y_K\}$ **do** // fine-tune $K$ models
    $f_c \leftarrow Initialize(\tilde{f})$
    $(U, c) \leftarrow \{(x, c) : x \in U\}$
    $f_c \leftarrow EarlyStoppedFineTuning\ (f_c, S \cup (U, c); V)$

**return** $\{f_{y_i}\}_{i=1}^K$

---

**Algorithm 2:** Novelty detection using ERD

**Input :** Ensemble $\{f_{y_i}\}_{i=1}^K$, Test set $T$, $O = \emptyset$, Threshold $t_0$, Disagreement metric $\rho$

**Result:** $O$, i.e. the novel-class samples from $T$

**for** $x \in T$ **do** // run hypothesis test
    **if** $(Avg \circ \rho)(f_{y_1}, ..., f_{y_K})(x) > t_0$ **then**
        $O \leftarrow O \cup \{x\}$

**return** $O$

---

validation accuracy is high and training error on $S \cup (U, c)$ is low. We create a diverse ensemble of $K$ classifiers $f_c$ by choosing a different artificial label $c \in \mathcal{Y}$ for every model.

Intuitively, encouraging each model in the ensemble to fit different labels to the unlabeled set $U$ promotes disagreement, as shown in Figure 2. In the next sections, we elaborate on how to use diverse ensembles for novelty detection.

## 2.2 ENSEMBLE DISAGREEMENT FOR NOVELTY DETECTION

We now discuss how we can use ensembles with disagreement to detect OOD samples and why the right amount of diversity is crucial. Note that we can cast the novelty detection problem as a hypothesis test with the null hypothesis $H_0 : x \in \mathcal{X}_{ID}$.

As usual, we test the null hypothesis by comparing a test statistic with a threshold $t_0$: The null hypothesis is *rejected* and we report $x$ as OOD (*positive*) if the test statistic is larger than $t_0$ (Section 4.3 elaborates on the choice of $t_0$). In particular, we use as test statistic the following disagreement score, which computes the average distance between the softmax outputs of the $K$ models in the ensemble:

$$(Avg \circ \rho)(f_1(x), ..., f_K(x)) := \frac{2\sum_{i \neq j} \rho(f_i(x), f_j(x))}{K(K-1)},$$

where $\rho$ is a measure of disagreement between the softmax outputs of two predictors, for example the total variation distance $\rho_{\text{TV}}(f_i(x), f_j(x)) = \frac{1}{2}\|f_i(x) - f_j(x)\|_1$ used in our experiments[2]. We provide a thorough discussion on the soundness of this test statistic for disagreeing models and compare it with previous metrics in Appendix B.

Even though previous work like Yu and Aizawa [2019] used a similar disagreement score, their detection performance is notably worse. The reason lies in the lack of diversity in their trained ensemble (see Figure 3a in Appendix B). On the other hand Algorithm 1 without early stopping would lead to a too diverse ensemble, that also disagrees on ID points, and hence, has a high false positive rate (see Appendix K). In

---

[2]We also expect other distance metrics to be similarly effective.

---

the next section, we explain why novelty detection with this test statistic crucially relies on the right amount of ensemble diversity and how ensembles may achieve this goal if they are trained to have regularized disagreement.

## 2.3 DESIRED ENSEMBLE DIVERSITY VIA REGULARIZED DISAGREEMENT

For simplicity of illustration, let us first assume a training set with binary labels and a semi-supervised novelty detection setting as depicted in Figure 2 a). For an ensemble with two models, like in Figure 2 b), the model predictions *agree* on the blue and red areas and *disagree* on the gray area depicted in Figure 2 c). Note that the two models in Figure 2 are *just diverse enough* to obtain both high power (flag true OOD as OOD) and low false positive rate (avoid flagging true ID as OOD) at the same time.

Previous methods that try to leverage unlabeled data to obtain more diverse ensembles either do not work with deep neural networks [Bennett et al., 2002, Zhang and Zhou, 2010, Jain et al., 2020] or do not disagree enough on OOD data [Yu and Aizawa, 2019], leading to subpar novelty detection performance (see Figure 3a in Appendix B).

To obtain the right amount of diversity, it is crucial to train ensembles with *regularized disagreement* on the unlabeled set: The models should disagree on the unlabeled OOD samples, but *agree* on the unlabeled ID points (Figure 3c). Thus, we avoid having too little disagreement as in Figure 3a), which results in low power, or too much diversity, resulting in high false positive rate as in Figure 3b). In particular, if models $f_c$ predict the correct label on ID points and the label $c$ on OOD data, we can effectively use disagreement to detect novel-class samples. Since classifiers with good ID generalization need to be smooth, we expect the model predictions on holdout OOD data from the same distributions to be in line with the predictions on the unlabeled set.

In Section 3 we argue that the training procedure in Algorithm 1 successfully induces *regularized disagreement* and prove it in a synthetic setting. Our experiments in Section 4 further corroborate our theoretical statements. Finally, we note that one could also use other regularization tech-

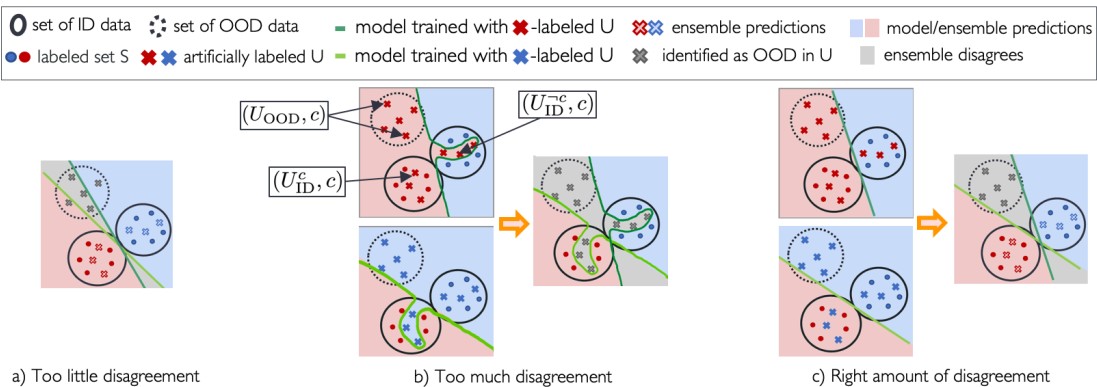

Figure 3: a) Ensembles with too little disagreement fail to detect OOD samples. b) An ensemble of two models trained on $S \cup (U, c)$ disagrees on both ID and OOD data. b) Regularization prevents models from fitting $(U_{\text{ID}}^{-c}, c)$, limiting disagreement to only OOD samples.

niques like dropout or weight decay. However, running a grid search to select the right hyperparameters can be more computationally expensive than simply using one run of the training process to select the optimal stopping time.

# 3 PROVABLE REGULARIZED DISAGREEMENT VIA EARLY STOPPING

In this section, we show how using early stopping in Algorithm 1 prevents fitting the incorrect artificial label on the unlabeled ID samples. Albeit for a simplified setting, this result provides a rigorous proof of concept and intuition for why ERD ensembles achieve the right amount of diversity necessary for good novelty detection.

## 3.1 PRELIMINARY DEFINITIONS

We first introduce necessary definitions to prepare the mathematical statement. Recall that in our approach, in addition to the correct labels of the ID training set $S$, each member of the ensemble tries to fit one label $c$ to the entire unlabeled set $U$ that can be further partitioned into

$$(U, c) = (U_{\text{ID}}, c) \cup (U_{\text{OOD}}, c)$$
$$= \{(x, c) : x \in U_{\text{ID}}\} \cup \{(x, c) : x \in U_{\text{OOD}}\},$$

where $U_{\text{ID}} := U \cap \mathcal{X}_{ID}$ and $U_{\text{OOD}} := U \setminus U_{\text{ID}}$. Moreover, assuming that the label of an ID input $x$ is deterministically given by $y^*(x)$, we can partition the set $(U_{\text{ID}}, c)$ (see Figure 3b) into a subset of effectively "correctly labeled" samples $(U_{\text{ID}}^c, c)$ and "incorrectly labeled" samples $(U_{\text{ID}}^{-c}, c)$:

$$(U_{\text{ID}}^{-c}, c) := \{(x, c) : x \in U_{\text{ID}} \text{ with } y^*(x) \neq c\}$$
$$(U_{\text{ID}}^c, c) := \{(x, c) : x \in U_{\text{ID}} \text{ with } y^*(x) = c\}.$$

Note that $(U_{\text{ID}}^{-c}, c)$ can be viewed as the subset of noisy samples from the entire training set $S \cup (U, c)$.

## 3.2 MAIN RESULT

We now prove that there exists indeed an optimal stopping time at which a two-layer neural network trained with gradient descent does not fit the incorrectly labeled subset $(U_{\text{ID}}^{-c}, c)$, under mild distributional assumptions.

For the formal statement, we assume that the artificially labeled set $S \cup (U, c)$ is *clusterable*, i.e. the points can be grouped in $K$ clusters of similar sizes. Each class may comprise several clusters, but every cluster contains only samples from one class. Any cluster may include at most a fraction $\eta \in [0, 1]$ of samples with label noise, e.g. $(U_{\text{ID}}^{-c}, c)$. We denote by $c_1, ..., c_K$ the cluster centers and define the matrix $C := [c_1, ..., c_K]^T \in \mathbb{R}^{K \times d}$. Further, let $\lambda_C^{\text{NN}}$ be a measure of how well a randomly-initialized two-layer neural network can separate the cluster centers. We provide the formal definition of $\lambda_C^{\text{NN}}$ in Appendix A. Intuitively, $\lambda_C^{\text{NN}}$ is large if the cluster centers are well-separated and it vanishes if $c_i = c_j$ for some $i, j \leq K$. Under these assumptions we have the following:

**Proposition 3.1.** *(informal) It holds with high probability over the initialization of the weights that a two-layer neural network trained on $S \cup (U, c)$ perfectly fits $S$, $(U_{\text{ID}}^c, c)$ and $(U_{\text{OOD}}, c)$, but not $(U_{\text{ID}}^{-c}, c)$, after $T \simeq \frac{\|C\|^2}{\lambda_C^{\text{NN}}}$ iterations.*

The precise assumptions for the proposition can be found in Appendix A. On a high level, the reasoning follows from two simple insights: 1. When the artificial label is not equal to the true label, the ID samples in the unlabeled set can be seen as noisy samples in the set $S \cup (U, c)$. 2. It is well known that early stopping prevents models from fitting incorrect labels since noisy samples with incorrect labels are often fit later during training (see e.g. theoretical and empirical evidence here Yilmaz and Heckel [2019], Li et al. [2020], Song et al. [2020], Liu et al. [2020]). In particular, our proof heavily relies on Theorem 2.2 of Li et al. [2020] which shows that early stopped predictors are robust to label noise.

Proposition 3.1 gives a flavor of the theoretical guarantees that ERD enjoys. Albeit simple, the clusterable data model actually includes data with non-linear decision boundaries. On the other hand, the requirement that the clusters are balanced seems rather restrictive. In our experiments we show that this condition is in fact more stringent than it should. In particular, our method still works when the number of OOD samples $|U_{\text{OOD}}|$ is considerably smaller than the number of ID samples from any given class, as we show in Section 4.5.

## 3.3 CHOOSING THE EARLY STOPPING TIME

In practice, we avoid computing the exact value of $T$ by using instead a heuristic for picking the early stopping iteration with the highest validation accuracy (indicated by the vertical line in Figure 4). As shown in the figure, the model fits the noisy training points, i.e. $(U_{\text{ID}}^{\neg c}, c)$, late during fine-tuning, which causes the validation accuracy to decrease, since the model will also predict the incorrect label $c$ on some validation ID samples. In Appendix J we show that the trend in Figure 4 is consistent across data sets.

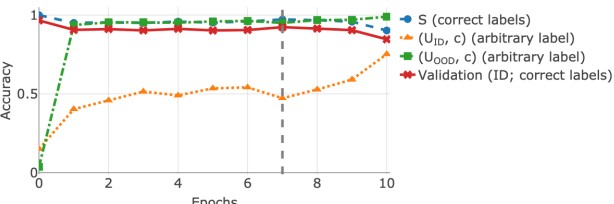

Figure 4: Accuracy during fine-tuning a model pretrained on $S$ (epoch 0 indicates values obtained with the initial pretrained weights). The samples in $(U_{\text{OOD}}, c)$ are fit first, while the model reaches high accuracy on $(U_{\text{ID}}, c)$ much later. We fine-tune for at least one epoch and then early stop when the validation accuracy starts decreasing after 7 epochs (vertical line). The model is trained on SVHN[0:4] as ID and SVHN[5:9] as OOD.

## 4 EXPERIMENTAL RESULTS

In this section we evaluate the novelty detection performance of ERD with deep neural networks on several image data sets. On difficult near OOD data sets, we find that our approach outperforms all baselines, including SSND methods, but also methods operating in other, sometimes more favorable settings. In addition, we discuss some of the trade-offs that impact ERD's performance.

### 4.1 DATA SETS

Our experiments focus on novel-class detection scenarios where the ID and OOD data share many similar features

and only differ in a few characteristics. We use standard image data sets (e.g. CIFAR10/CIFAR100) and consider half of the classes as ID, and the other half as novel. We also assess ERD's performance on a medical image benchmark [Cao et al., 2020], where near OOD data consists of novel unseen diseases (e.g. X-rays of the same body part from patients with different conditions; see Appendix E for details). Further, we also include far OOD data sets (e.g. CIFAR10/CIFAR100 vs SVHN) for completeness.

For all scenarios, we used a labeled training set (e.g. 40K samples for CIFAR10), a validation set with ID samples (e.g. 10K samples for CIFAR10) and an unlabeled test set where half of the samples are ID and the other half are OOD (e.g. 5K ID samples and 5K OOD samples for CIFAR10 vs SVHN). For evaluation, we use a holdout set containing ID and OOD samples in the same proportions as the unlabeled set. Moreover, in Appendix F.5 we present results obtained with a smaller unlabeled set of only 1K samples.

### 4.2 BASELINES

We compare our method against a wide range of baselines that are applicable in the SSND setting.

**Semi-supervised novelty detection.** We primarily compare ERD to SSND approaches that are designed to incorporate a small set of unlabeled ID and novel samples.

The *MCD* method [Yu and Aizawa, 2019] trains an ensemble of two classifiers such that one model gives high-entropy and the other yields low entropy predictive distributions on the unlabeled samples. Furthermore, *nnPU* [Kiryo et al., 2017] considers a binary classification setting, in which the labeled data comes from one class (i.e. ID samples, in our case), while the unlabeled set contains a mixture of samples from both classes. Notably, both methods require oracle knowledge that is usually unknown in the regular SSND setting: MCD uses test OOD data for hyperparameter tuning while nnPU requires oracle knowledge of the ratio of OOD samples in the unlabeled set.

In addition to these baselines, we also propose two natural extensions to the SSND setting of two existing methods. Firstly, we present a version of the Mahalanobis approach (*Mahal-U*) that is calibrated using the unlabeled set, instead of using oracle OOD data. Secondly, since nnPU requires access to the OOD ratio of the unlabeled set, we also consider a less burdensome alternative: a *binary classifier* trained to separate the training data from the unlabeled set and regularized with early stopping like our method.

**Unsupervised novelty detection (UND).** Naturally, one may ignore the unlabeled data and use unsupervised approaches. The current SOTA UND method on the usual benchmarks is the *Gram method* [Sastry and Oore, 2019]. Other notable UND approaches include *vanilla ensembles*

Table 1: AUROC and TNR@95 for ERD and various baselines (we *highlight* the best method for each data set). Numbers in square brackets indicate the ID/OOD classes. Asterisks mark methods proposed in this paper. Mahal, nnPU and MCD ([†]) use oracle information about the OOD data. Repeated runs of ERD show a small variance $\sigma^2 < 0.01$ in the detection metrics.

| ID data | OOD data | Other settings | | | | | SSND | | | | |
|---|---|---|---|---|---|---|---|---|---|---|---|
| | | Vanilla Ensembles | Gram | DPN | OE | Mahal.[†] | nnPU[†] | MCD[†] | Mahal-U | Bin. Classif. * | ERD * |
| | | AUROC ↑ / TNR@95 ↑ | | | | | | | | | |
| SVHN | CIFAR10 | 0.97 / 0.88 | 0.97 / 0.86 | *1.00 / 1.00* | *1.00 / 1.00* | 0.99 / 0.98 | *1.00 / 1.00* | 0.97 / 0.85 | 0.99 / 0.95 | 1.00 / 1.00 | 0.99 / 0.98 |
| CIFAR10 | SVHN | 0.92 / 0.78 | *1.00 / 0.98* | 0.95 / 0.85 | 0.97 / 0.89 | 0.99 / 0.96 | *1.00 / 1.00* | *1.00 / 0.98* | 0.99 / 0.96 | 1.00 / 1.00 | *1.00 / 1.00* |
| CIFAR100 | SVHN | 0.84 / 0.48 | 0.99 / 0.97 | 0.77 / 0.44 | 0.82 / 0.50 | 0.98 / 0.90 | *1.00 / 1.00* | 0.97 / 0.73 | 0.98 / 0.92 | 1.00 / 1.00 | *1.00 / 1.00* |
| SVHN [0:4] | SVHN [5:9] | 0.92 / 0.69 | 0.81 / 0.31 | 0.87 / 0.19 | 0.85 / 0.52 | 0.92 / 0.71 | *0.96 / 0.73* | 0.91 / 0.51 | 0.91 / 0.63 | 0.81 / 0.40 | 0.95 / *0.73* |
| CIFAR10 [0:4] | CIFAR10 [5:9] | 0.80 / 0.39 | 0.67 / 0.15 | 0.82 / 0.32 | 0.82 / 0.41 | 0.79 / 0.27 | 0.61 / 0.11 | 0.69 / 0.25 | 0.64 / 0.13 | 0.85 / 0.43 | *0.89 / 0.57* |
| CIFAR100 [0:49] | CIFAR100 [50:99] | 0.78 / 0.35 | 0.71 / 0.16 | 0.70 / 0.26 | 0.74 / 0.31 | 0.72 / 0.20 | 0.53 / 0.06 | 0.70 / 0.26 | 0.72 / 0.19 | 0.66 / 0.13 | *0.81 / 0.41* |

[Lakshminarayanan et al., 2017], deep generative models (which tend to give undesirable results for OOD detection [Kirichenko et al., 2020]), or various Bayesian approaches (which are often poorly calibrated on OOD data [Ovadia et al., 2019]).

Preliminary analyses revealed that generative models and methods trained with a contrastive loss [Winkens et al., 2020] or with one-class classification [Sohn et al., 2021] perform poorly on near OOD data sets (see Appendix F.2 for a comparison; we use numbers reported by the authors for works where we could not replicate their results).

**Other methods.** We also compare with *Outlier Exposure* [Hendrycks et al., 2019] and *Deep Prior Networks (DPN)* [Malinin and Gales, 2018] which use TinyImages as known outliers during training, irrespective of the OOD set used for evaluation. On the other hand, the *Mahalanobis* baseline [Lee et al., 2018] is tuned on samples from the same OOD distribution used for evaluation. Finally, we also consider large transformer models pretrained on ImageNet21k and fine-tuned on the ID training set [Fort et al., 2021].

### 4.3 IMPLEMENTATION DETAILS

**Baseline hyperparameters.** For all the baselines, we use the default hyperparameters suggested by their authors on the respective ID data set (see Appendix D for more details). For the binary classifier, nnPU, ViT, and vanilla ensembles, we choose the hyperparameters that optimize the loss on an ID validation set.

**ERD details.** [3] We follow the procedure in Algorithm 1 to fine-tune each model in the ERD ensemble starting from weights that are pretrained on the labeled ID set $S$.[4] Unless

---

[3] Our code is publicly available at https://github.com/ericpts/ERD.

[4] In the appendix we also train the models from random initializations, i.e. ERD++, and obtain better novelty detection at the cost of more training iterations.

otherwise specified, we train $K = 3$ ResNet20 networks [He et al., 2016] using 3 randomly chosen class labels for $(U, c)$ and note that even ensembles of two models produce good results (see Appendix F.9). We stress that whenever applicable, our choices disadvantage ERD for the comparison with the baselines, e.g. vanilla ensembles use $K = 5$, and for most of the other approaches we use the larger WideResNet-28-10. We select the early stopping time and other standard hyperparameters so as to maximize validation accuracy.

**Evaluation.** As in standard hypothesis testing, choosing different thresholds for rejecting the null hypothesis leads to different false positive and true positive rates (FPR and TPR, respectively). The ROC curve follows the FPR and TPR for all possible threshold values and the area under the curve (AUROC; larger values are better) captures the performance of a statistical test without having to select a specific threshold. In addition, we also report the TNR at a TPR of 95% (TNR@95; larger values are better). These metrics evaluate the quality of an outlier score without choosing a rejection threshold. However, we note that this problem can easily be addressed in practice. For instance, one can choose the threshold so as to achieve a desired FPR, which can be estimated using a validation set of ID samples.[5]

**Computation cost.** We only need to fine-tune two-model ensembles to get good performance with ERD (see Appendix F.9). For instance, in applications like the one in Figure 1, ERD fine-tuning introduces little overhead and works well even with scarce resources (e.g. it takes around 5 minutes on 2 GPUs for the settings in Table 1). In contrast, other ensemble diversification methods require training different models for each hyperparameter choice and have training losses that cannot be easily parallelized (e.g. Yu and Aizawa [2019]). Moreover, the only other approach that achieves comparable performance to our method on *some*

---

[5] Alternatively, the work of [Liu et al., 2018] proposes a criterion for selecting the threshold, tailored specifically to the SSND setting. This method uses the unlabeled set and the known ID data to estimate the distribution of outlier scores for OOD points.

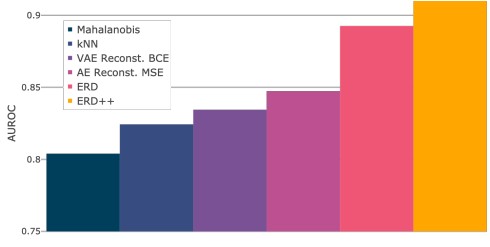
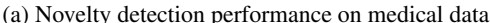

(a) Novelty detection performance on medical data

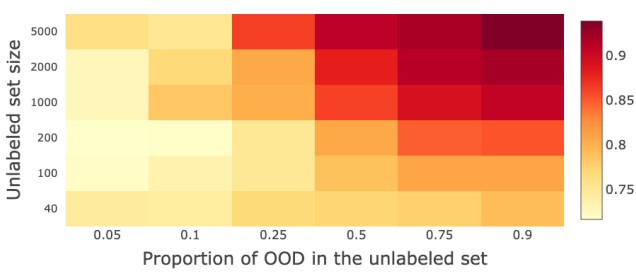

(b) Effect of OOD proportion on detection

Figure 5: **Left:** AUROC averaged over all scenarios in the medical novelty detection benchmark. The values for the baselines are computed using the code from Cao et al. [2020]. **Right:** The AUROC of a 3-model ERD ensemble as the number and proportion of ID (CIFAR10[0:4]) and OOD (CIFAR10[5:9]) samples in the unlabeled set are varied (see also Appendix I).

near OOD data uses large transformer models pretrained on a large and conveniently chosen data set [Fort et al., 2021].

### 4.4 MAIN RESULTS

We summarize the main empirical results in Table 1. While most methods achieve near-perfect detection for far OOD, ERD has a clear edge over the baselines for novel-class detection within the same dataset – even compared to methods (†) that use oracle OOD information. For completeness, we present in Appendix F.2 a comparison with more related works. These methods either show unsatisfactory performance on near OOD tasks, or seem to work well only on certain specific data sets. We elaborate on the potential causes of failure for these works in Section 5.

For the medical novelty detection benchmark we show in Figure 5a the average AUROC achieved by some representative baselines taken from Cao et al. [2020]. Our method improves the average AUROC from $0.85$ to $0.91$, compared to the best baseline. We refer the reader to Cao et al. [2020] for precise details on the methods. Appendix G contains more results, as well as additional baselines.

### 4.5 ABLATION STUDIES AND LIMITATIONS

We also perform extensive experiments to understand the importance of specific design choices and hyperparameters, and refer the reader to the appendix for details.

**Relaxing assumptions on OOD samples.** In Table 1 we evaluate our approach on a holdout test set that is drawn from the same distribution as the unlabeled set $U$ used for fine-tuning. However, we provide experiments in Appendix F.10 that show that novelty detection with ERD continues to perform well even when the test set and $U$ come from different distributions (e.g. novel-class data in the test set also suffers from corruptions). Further, even though our main focus is novel-class detection, our experiments (Ap-

pendix F.4) indicate that ERD can also successfully identify near OOD samples that suffer from only mild covariate shift compared to the ID data (e.g. CIFAR10 vs corrupted CIFAR10 [Hendrycks and Dietterich, 2019] or CIFAR10v2 [Recht et al., 2019]). Finally, Appendix F.1 shows that ERD ensembles also perform well in a transductive setting [Scott and Blanchard, 2008], where the test set coincides with $U$.

**Relaxing the assumptions of Proposition 3.1.** Our theoretical results require that the ID classes in the training set and the novel classes in $U$ have similar cardinality. In fact, this condition is unnecessarily strong as we show in our empirical analysis: In all experimental settings we have significantly fewer OOD than ID training points. We further investigate the impact of the size of the unlabeled set and of the ratio of novel samples in it ($\frac{|U_{\text{OOD}}|}{|U_{\text{ID}}|+|U_{\text{OOD}}|}$) and find that ERD in fact maintains good performance for a broad range of ratios in Figure 5b.

**Sensitivity to hyperparameter choices.** We point out that ERD ensembles are particularly robust to changes in the hyperparameters like batch size or learning rate (Appendix H), or the choice of the arbitrary label assigned to the unlabeled set (Appendix F.9). Further, we note that ERD ensembles with as few as two models already show remarkable novelty detection performance and refer to Appendix F.9 for experiments with larger ensemble sizes. Moreover, ERD performance improves with larger neural networks (Appendix F.8), meaning that ERD will benefit from any future advances in architecture design.

**Choice of disagreement score.** We show in Table 1 in Appendix B, that the training procedure alone (Algorithm 1) does not suffice for good novelty detection. For optimal results, ERD ensembles need to be combined with a disagreement-based score like the one introduced in Section 2.3. Finally, we show how the distribution of the disagreement score changes during training for ERD (Appendix K) and explain why regularizing disagreement is more challenging for near OOD data, compared to easier, far OOD settings (Appendix J).

Table 2: Taxonomy of novelty detection methods, categorized according to data availability (**horizontal axis**) and probabilistic perspective (**vertical axis**). We highlight the ensemble-based methods.

| | UND | SSND | Different OOD A-UND | Synthetic OOD A-UND | P-UND | SND |
|---|---|---|---|---|---|---|
| Learn $P_X$ | Generative e.g. [AAB18], OC classif. e.g. [SPSSW01] | nnPU [KNPS17] | | OC classif. [SLYJP21, TMJS20] | [RH21] | [GKRB13, DKT19, RVGBM20] |
| Learn $P_X$ using $y$ | Gram [SO19], OpenHybrid [ZLGG20] | **ERD (Ours)**, SSND for shallow models [MBGBM10, BLS10], U-LAC [DYZ14, ZZMZ20] | | Data augmentation for contrastive loss [TMJS20, LA20] | ViT [FRL20] | Mahala. [LLLS18], **MCD [YA19]** |
| Uncertainty of $P_{Y\mid X}$ | Bayesian methods e.g. [GG16], **Vanilla Ensemble [LPB17]** | — | DPN [MG18], OE [HMD19] | GAN outputs [LLLS18], noise [HTLI19] or uniform samples (**[JLMG20]**) | | ODIN [LLS18] |

**Limitations.** Despite the advantages of ERD, like all prior SSND methods, our approach is not a good fit for online (real-time) novelty detection tasks. Moreover, ERD ensembles are not tailored to anomaly detection, where outliers are particularly rare, since the unlabeled set should contain at least a small number of samples from the novel classes (see Figure 5b and Appendix I). However, ERD ensembles are an ideal candidate for applications that require highly accurate, offline novelty detection, like the one illustrated in Figure 1.

## 5 RELATED WORK

In this section, we present an overview of different types of related methods that are in principle applicable for solving semi-supervised novelty detection. In particular, we indicate caveats of these methods based on their categorization with respect to 1) data availability and 2) the surrogate objective they try to optimize. This taxonomy may also be of independent interest to navigate the zoo of ND methods. We list a few representative approaches in Table 2 and refer the reader to surveys such as Bulusu et al. [2020] for a thorough literature overview.

### 5.1 TAXONOMY ACCORDING TO DATA AVAILABILITY

In this section we present related novelty detection methods that use varying degrees of labeled OOD data for training. We call *test OOD* the novel-class data that we want to detect at test time.

In a scenario like the one in Figure 1, one can apply ***unsupervised novelty detection (UND)*** methods that ignore the unlabeled batch and only uses ID data during training [Lakshminarayanan et al., 2017, Sastry and Oore, 2019, Nalisnick et al., 2019]. However, these approaches lead to poor novelty detection performance, especially on near OOD data.

There are methods that suggest to improve UND perfor-mance by using additional data. For example, during training one may use synthetically generated outliers (e.g. Tack et al. [2020], Sohn et al. [2021]) or a different OOD data set that may be available (e.g. OE and DPN use TinyImages) with samples *known to be outliers*. However, in order for these ***augmented unsupervised ND (A-UND)*** methods to work, they require that the OOD data used for training is similar to test OOD samples. When this condition is not satisfied, A-UND performance deteriorates drastically (see Table 1). However, by definition, novel data is unknown and the only information about the OOD data that is realistically available is in the unlabeled set like in SSND. Therefore, it is unknown what an appropriate choice of the training OOD data is for A-UND methods.

Another line of work uses pretrained models to incorporate additional data that is close to test OOD samples, i.e. ***pretrained UND (P-UND)***. Fort et al. [2021] use large transformer models pretrained on ImageNet21k and achieve good near OOD detection performance when ID and OOD data are similar to ImageNet samples (e.g. CIFAR10/CIFAR100). However, our experiments in Appendix F.3 reveal that this method performs poorly on all other near OOD data sets, including unseen FashionMNIST or SVHN classes and X-rays of unknown diseases. This unsatisfactory performance is apparent when ID and OOD data do not share visual features with the pretraining data (i.e. ImageNet21k). Since collecting such large troves of "similar" data for pre-training is often not possible in practical applications (as medical imaging), the use case of their method is rather limited.

Furthermore, a few popular methods use test OOD data for calibration or hyperparameter tuning [Yu and Aizawa, 2019, Lee et al., 2018, Liang et al., 2018, Ruff et al., 2020], which is not applicable in practice. Clearly, knowing the test OOD distribution a priori turns the problem into ***supervised ND (SND)***, and hence, violates the fundamental assumption that OOD data is unforeseeable.

As we have already seen, current ***SSND*** approaches (e.g. MCD, nnPU) perform poorly for complex models such as neural networks. We note that SSND is similar to using unlabeled data for learning with augmented classes (U-LAC)

[Da et al., 2014, Guo et al., 2020, Zhang et al., 2020] and is related to transductive novelty detection [Scott and Blanchard, 2008, Guo et al., 2020], where the test set coincides with the unlabeled set used for training.

## 5.2 TAXONOMY ACCORDING TO PROBABILISTIC PERSPECTIVE

Apart from data availability, the methods that we can use in a practical SSND scenario implicitly or explicitly use a different principle based on a probabilistic model. For example, novel-class samples are a subset of the points that are out-of-distribution in the literal sense, i.e. $P_X(x) < \alpha$. One can hence **learn** $P_X$ from unlabeled ID data, which is however notoriously difficult in high dimensions.

Similarly, from a Bayesian viewpoint, the predictive variance is larger for OOD samples with $P_X(x) < \alpha$. Hence, one could instead compute the posterior $P_X(y|x)$ and flag points with large variance (i.e. high **predictive uncertainty**). This circumvents the problem with estimating $P_X$. However, Bayesian estimates of uncertainty that accompany NN predictions tend to not be accurate on OOD data [Ovadia et al., 2019], resulting in poor novelty detection performance.

When the labels are available for the training set, we can instead partially **learn** $P_X$ **using** $y$. For instance, one could use generative modeling to estimate the set of $x$ for which $P_X(x) > \alpha$ via $P_X(x|y)$ Lee et al. [2018], Sastry and Oore [2019]. Alternatively, given a loss and function space, we may use the labels indirectly, like in ERD, and use properties of the approximated population error that imply small or large $P_X$.

## 6 CONCLUSION

In summary, we propose an SSND procedure that exploits unlabeled data effectively to generate an ensemble with *regularized* disagreement, which achieves remarkable novelty detection performance. Our SSND method does not need labeled OOD data during training unlike many other related works summarized in Table 2.

We leave as future work a thorough investigation of the impact of the labeling scheme of the unlabeled set on the sample complexity of the method, as well as an analysis of the trade-off governed by the complexity of the model class.

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
