# OpenReview forum: "Semi-supervised novelty detection using ensembles with regularized disagreement"
_auai.org/UAI/2022/Conference — UAI 2022 Poster_

### Official Review · Reviewer_vUMd · 2022-03-30

**Q2(1) Originality/Novelty:** 3
**Q2(2) Significance/Impact:** 3
**Q2(3) Correctness/Technical Quality:** 3
**Q2(6) Clarity Of Writing:** 2
**Q6 Overall Score:** 6
**Q8 Confidence In Your Score:** 2

**Q1 Summary And Contributions:**

This paper proposes a novelty detection method with semi-supervision, where a few labeled inlier data are available.
By leveraging the idea that class predictions for novelty data should be "diverse", the proposed method trains a classifier given with supervised inlier data and unlabeled data labeled with a specific class and aggregates the classifiers to see how diverse the outputs are.
The thorough empirical evaluation witnesses that the proposed method outperforms many existing methods.

**Q2 Assessment Of The Paper:**

More detailed information regarding each of these aspects is given below:

**Q2(4) Quality Of Experiments (Optional):**

4: Excellent: The experimental evaluation is comprehensive and the results are compelling.

**Q2(5) Reproducibility:**

4: Excellent: Key resources (e.g., proofs, code, data) are available and key details (e.g., proof sketches, experimental setup) are comprehensively described for competent researchers to confidently and easily reproduce the main results.

**Q3 Main Strengths:**

The empirical performance of the proposed method is significant, compared with several recent baselines for novelty detection.
The authors also validated the usefulness of the proposed method with the medical dataset.

The proposed method is based on the simple idea that novelty data should have diverse outputs with respect to class-specific classifiers.
This belief is intuitive and indeed works very well.

**Q4 Main Weakness:**

Although the authors provide a theoretical result (Prop. 3.1) showing that there exists an optimal number of the iteration to prevent the model from fitting the noisy inputs, they opt to use validation accuracy to choose the number of iterations.
This somewhat undermines the usefulness of the theoretical result.
Indeed, the optimal $T$ given in Prop. 3.1 does not seem easy to know in practice.

The proposed method relies on hypothesis testing, by looking at whether the test statistic (evaluating the diversity of predictions) is above a given threshold.
In the experiments, the authors use the ROC to evaluate the performance, so it is not a big issue for the model to have a threshold parameter.
However, in practice, we need to determine the threshold to control TPR/FPR appropriately beforehand, which would not be straightforward under the semi-supervised novelty detection setting.

**Q5 Detailed Comments To The Authors:**

- Can the authors elaborate on the difference between novelty detection and anomaly detection? Is it a matter of the ratio of novelty or anomaly? In addition, I would like to know what "singular" does mean in the second line from the end of p.1 right column.

- In Prop. 3.1, it is not clear what $\lambda_C^{\mathrm{NN}}$ means. Also, the superscript should be in roman $\mathrm{NN}$. If a little bit more details are provided in the main part, it must help readers to understand.

**Q7 Justification For Your Score:**

While the proposed method has an issue in threshold parameter selection, the empirical performance is great in itself.
Hence, I am inclined to accept this paper.

**Q9 Complying With Reviewing Instructions:**

1: Yes.

---

### Official Review · Reviewer_PJMk · 2022-04-12

**Q2(1) Originality/Novelty:** 4
**Q2(2) Significance/Impact:** 3
**Q2(3) Correctness/Technical Quality:** 3
**Q2(6) Clarity Of Writing:** 3
**Q6 Overall Score:** 8
**Q8 Confidence In Your Score:** 3

**Q1 Summary And Contributions:**

The paper proposes an out-of-distribution detection method, by training an ensemble of classifiers.
The classifiers are trained to predict different labels for an unlabeled part of the datsets, which is a mixture of in distribution and out-of-distribution data.
With early stopping the classifiers still predict the correct labels for in-distribution data, but they will predict different labels for OOD data, which allows the OOD data to be detected.

**Q2 Assessment Of The Paper:**

More detailed information regarding each of these aspects is given below:

**Q2(4) Quality Of Experiments (Optional):**

3: Good: The experimental evaluation is adequate, and the results convincingly support the main claims.

**Q2(5) Reproducibility:**

3: Good: Key resources (e.g., proofs, code, data) are available and key details (e.g., proofs, experimental setup) are sufficiently well-described for competent researchers to confidently reproduce the main results.

**Q3 Main Strengths:**

* The intuitive explanation makes sense
* The figures clearly illustrate the idea behind the method
* There is an analysis of the sensitivity to the early stopping method
* Lots of details are included in the appendix

**Q4 Main Weakness:**

* The experiments are not described in enough detail in the main paper.
* The presentation of the results is not very clear.
* The method relies heavily on a validation set, this is not discussed much.

**Q5 Detailed Comments To The Authors:**

* Table 1 is hard to read, because it combines two different metrics, maybe use two tables instead?
* Marking the results of ERD in blue and bold makes it look like the best result, even when it is not. Just only highlight the best among both the baselines and the new method.
* Table 1 could use less whitespace between the methods, allowing for a larger font.

* Figure 5a: put the algorithms on the horizontal axis instead of a legend.

* "Any cluster may include at most a fraction ρ"
  This is reusing the symbol \rho, which previously denoted a measure of disagreement.

* Algorithm 2 is better expressed as a set comprehension


**Q7 Justification For Your Score:**

This is an interesting new approach to the problem, with a clear explanation and with good results.

**Q9 Complying With Reviewing Instructions:**

1: Yes.

---

### Official Review · Reviewer_x5jD · 2022-04-13

**Q2(1) Originality/Novelty:** 2
**Q2(2) Significance/Impact:** 3
**Q2(3) Correctness/Technical Quality:** 2
**Q2(6) Clarity Of Writing:** 3
**Q6 Overall Score:** 5
**Q8 Confidence In Your Score:** 4

**Q1 Summary And Contributions:**

This paper proposes to find a semi-supervised novelty detection method which finds ensembles with diverse predictions. An early stopping strategy is also adopted to achieve regularized disagreement for some simplified assumptions. Experiments on benchmark datasets shows the effectiveness of the proposed method on novelty detection.

**Q2 Assessment Of The Paper:**

More detailed information regarding each of these aspects is given below:

**Q2(4) Quality Of Experiments (Optional):**

3: Good: The experimental evaluation is adequate, and the results convincingly support the main claims.

**Q2(5) Reproducibility:**

3: Good: Key resources (e.g., proofs, code, data) are available and key details (e.g., proofs, experimental setup) are sufficiently well-described for competent researchers to confidently reproduce the main results.

**Q3 Main Strengths:**

1. The proposed method is generally simple and reasonable. The description on the ensemble construction and early stopping is clear.
2. The experiments  are intensive.

**Q4 Main Weakness:**

1. The novelty and contribution are limited. The ensemble and early stopping ideas are not very creative for the problem in this work.
2. The optimality of the early stopping strategy should be proved. The stability of ensembles should be discussed.
3. The settings in the experiments are not normal, which should be explained in detail.

**Q5 Detailed Comments To The Authors:**

Many recent related researches on near-OOD and far-OOD detection have achieved good results, so the authors should emphasize what challenging issue this paper has addressed compared to existing works.
The presentation should be improved. E.g., the annotation in Fig. 9(a) may be wrong.

**Q7 Justification For Your Score:**

Very simple and effective method for novelty detection;
Limited novelty and contribution;
The fairness of the experimental setting is problematic;
Some presentation issues.

**Q9 Complying With Reviewing Instructions:**

1: Yes.

---

### Decision · Program_Chairs · 2022-05-15

**Decision:**

Accept (Poster)

**Comment:**

Meta Review: All reviewers have positive views on the experimental studies performed in this paper, which are sufficient to validate the effectiveness of the proposed approach. Furthermore, the idea presented in this paper is intuitive and the resulting approach is simple yet effective.

Several technical details can be made clearer in the revised version.